# β11-12 linker isomerization governs acid-sensing ion channel desensitization and recovery

**Matthew L Rook[1], Abby Williamson[2], John D Lueck[3], Maria Musgaard[4], David M Maclean[3]\***

[1]Graduate Program in Cellular and Molecular Pharmacology and Physiology, Rochester, United States; [2]Biomedical Engineering Program, University of Rochester, New York, United States; [3]Department of Pharmacology and Physiology, University of Rochester Medical Center, Rochester, United States; [4]Department of Chemistry and Biomolecular Sciences, University of Ottawa, Ottawa, Canada

**Abstract** Acid-sensing ion channels (ASICs) are neuronal sodium-selective channels activated by reductions in extracellular pH. Structures of the three presumptive functional states, high-pH resting, low-pH desensitized, and toxin-stabilized open, have all been solved for chicken ASIC1. These structures, along with prior functional data, suggest that the isomerization or flipping of the β11–12 linker in the extracellular, ligand-binding domain is an integral component of the desensitization process. To test this, we combined fast perfusion electrophysiology, molecular dynamics simulations and state-dependent non-canonical amino acid cross-linking. We find that both desensitization and recovery can be accelerated by orders of magnitude by mutating resides in this linker or the surrounding region. Furthermore, desensitization can be suppressed by trapping the linker in the resting state, indicating that isomerization of the β11–12 linker is not merely a consequence of, but a necessity for the desensitization process in ASICs.

**\*For correspondence:**
David_MacLean@urmc.rochester.edu

**Competing interests:** The authors declare that no competing interests exist.

## Introduction

Acid-sensing ion channels (ASICs) are a family of sodium-selective trimeric ion channels activated by extracellular acidification. This family is composed of four genes (five in humans) giving rise to six proton-sensitive isoforms which each have their own distinct expression profiles and biophysical properties (*Gründer and Pusch, 2015*; *Kellenberger and Schild, 2015*). Genetic studies have uncovered a variety of roles for ASICs such as in ischemic stroke, visceral pain sensation, epilepsy, substance abuse and fear conditioning (*Kellenberger and Schild, 2015*; *Lin et al., 2015*). As such, ASICs are attractive drug targets and there is considerable interest in understanding the structural basis for channel gating.

At physiological pH, ASICs are primarily found in a resting conformation. A rapid drop in extracellular pH triggers ASIC activation and desensitization, occurring over several milliseconds and hundreds of milliseconds, respectively (*Du et al., 2014*; *Kreple et al., 2014*; *MacLean and Jayaraman, 2016*; *MacLean and Jayaraman, 2017*; *Wemmie et al., 2008*). Proposed structures for each of the resting, open and desensitized states have been solved by X-ray crystallography for the chicken ASIC1 isoform (cASIC1) (*Baconguis et al., 2014*; *Baconguis and Gouaux, 2012*; *Gonzales et al., 2009*; *Jasti et al., 2007*; *Yoder et al., 2018*). These structural studies reveal ASICs to be trimers with each subunit consisting of short intracellular N and C termini, two transmembrane helices and a large extracellular domain. The extracellular domain (ECD) has been likened to a hand, with finger, knuckle, thumb, palm and β-ball domains (*Figure 1A*; *Jasti et al., 2007*). Interestingly, in the open and desensitized structures, the upper half of the ECD is nearly identical while the lower half and the

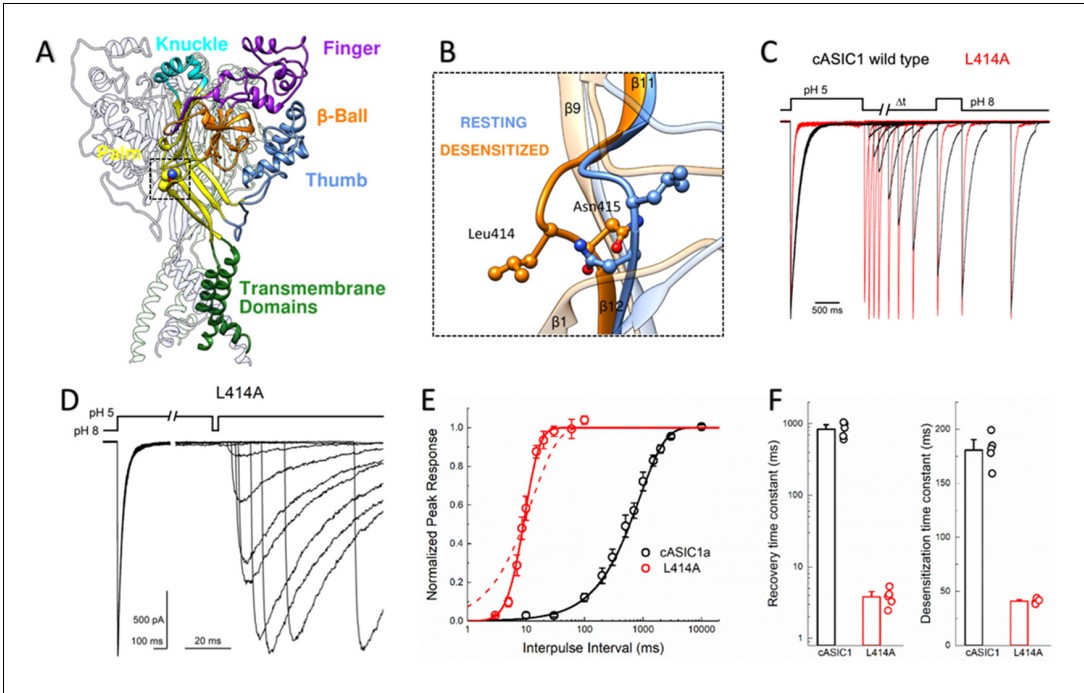

**Figure 1.** L414A drastically accelerates cASIC1 desensitization and recovery. (**A**) Crystal structure of the low-pH desensitized states of cASIC (*PDB:4NYK*) with major domains of one subunit colored and labeled. Boxed region contains the β11–12 linker. (**B**) Closeup view of locally aligned β11–12 linkers of resting (*blue, PDB:5WKV*) and desensitized states (*orange*). Leu414 and Asn415 side chains are depicted as ball and stick. The view has been rotated approximately 90 degrees compared to **A**). (**C**) Normalized outside-out patch recordings of wild-type cASIC1 (*black traces*) and L414A (*red traces*) during a paired pulse recovery protocol. (**D**) Representative recording of L414A responses during a modified paired pulse protocol to examine very brief inter-pulse intervals. Note the different time scales before and after the x-axis break. The pH protocol for the shortest inter-pulse interval of 3 ms is shown. (**E**) Recovery from desensitization time course across patches for cASIC1 wild type (*black*) and L414A (*red*). Solid lines are fits using *Equation 1*. The dashed line is a fit using *Equation 1* but with the slope factor, m, set to 1. (**F**) Summary of time constants of recovery from (*left panel*) and entry to (*right panel*) desensitization for wild type (*black bars and circles*) and L414A (*red bars and circles*). The error bars are S.E.M. and the circles are individual patches.

The online version of this article includes the following figure supplement(s) for figure 1:

**Figure supplement 1.** L414A does not substantially alter the pH dependence of activation.

transmembrane domains show substantial differences (*Baconguis and Gouaux, 2012*; *Yoder et al., 2018*). However, in the resting and desensitized structures, the lower ECD and transmembrane domain conformations are similar while the upper ECD is distinct. The transition zone between the upper ECD, similar in open and desensitized states, and the lower ECD, similar in resting and desensitized states, is marked by the linker between β-strands 11 and 12. Within this linker, Leu414 and Asn415 (chicken numbering) are known to be particularly important as mutations to these highly conserved residues have been reported to alter desensitization kinetics and activation curves (*Li et al., 2010a*; *Roy et al., 2013*; *Springauf et al., 2011*; *Wu et al., 2019*). These two residues undergo a dramatic motion between the open and desensitized states of the channel. In open (and resting) structures, the side chain of Leu414 is oriented outwards, away from the central axis of the channel, and forms a hydrophobic contact with Leu86 (*Baconguis et al., 2014*; *Baconguis and Gouaux, 2012*; *Yoder et al., 2018*). Asn415 is pointed downwards, toward the lower palm domain. However, in the desensitized state the residues swap orientations, with Leu414 pointing downwards and Asn415 swinging up (*Figure 1B*; *Gonzales et al., 2009*; *Jasti et al., 2007*). This substantial motion, as well as past functional data, has prompted the suggestion that this linker acts as a 'molecular clutch', coupling the conformational changes in the acidic pocket to the TMD, driving activation and subsequently disengaging during desensitization, enabling the upper ECD to maintain a

protonated conformation yet simultaneously allowing the lower ECD and TMDs to collapse and adopt a resting-like conformation (*Baconguis and Gouaux, 2012*; *Yoder et al., 2018*). Here, we set out to investigate the contribution of this linker, in particular Leu414, to the kinetics of both entry to and exit from desensitization using a combination of fast perfusion electrophysiology, molecular dynamics simulations and non-canonical amino acid UV-crosslinking.

## Results

### Leu414 strongly influences entry to and recovery from desensitization

The large extracellular domain of individual ASIC subunits has been likened to a hand shape with distinct thumb, finger, knuckle and palm domains (*Figure 1A*). The resting, open and desensitized state structures have been solved. These have revealed that within the palm domain, the linker connecting the β11 and β12 strands undergoes a substantial reorientation (*Baconguis et al., 2014*; *Baconguis and Gouaux, 2012*; *Gonzales et al., 2009*; *Yoder et al., 2018*). As seen in *Figure 1B*, in the resting state as well as in the toxin-stabilized open state, the side chain of Leu414 is oriented upwards and away from the central axis of the channel while Asn415 is pointing down and inwards. However, in the desensitized structures these amino acid residues undergo a 180-degree flip, essentially exchanging positions with Leu414 pointing downward and Asn415 pointing outward (*Figure 1B*). We hypothesized that this flip is an integral component of the desensitization process and makes a substantial contribution to the energy barrier separating the resting and desensitized states. Therefore, increasing the probability of linker 'flipping' should accelerate the entry to and exit from desensitization. While mutations to this linker have been previously reported to alter the rates into desensitization (*Li et al., 2010a*; *Roy et al., 2013*; *Springauf et al., 2011*; *Wu et al., 2019*) and the pH dependence of activation or steady-state desensitization, no study has examined their impact on the reverse process of recovery. Indeed, ASIC recovery from desensitization has been mechanistically examined rarely in general (*Kusama et al., 2013*; *Li et al., 2012*; *MacLean and Jayaraman, 2016*). Therefore, we began testing this hypothesis by mutating Leu414 to Ala, decreasing side chain size to reduce steric hindrance during the 'flipping motion' and examined both entry to and exit from desensitization. To do this, we employed a paired pulse protocol where an outside-out patch expressing cASIC1 was incubated at pH 8 to maximally populate the resting state, followed by a jump for 1.5 s into pH 5 to fully desensitize the channel population. Following this conditioning pulse, the patch was exposed to pH 8 again for variable intervals, ranging from 3 ms to 30 s, to enable some fraction of channels to recover before a 500 ms test pulse of pH 5 was applied (*Figure 1C*). A ratio of the second peak to the first enabled us to determine the fraction of the response recovered as a function of the interval between the end of the conditioning pulse and the beginning of the test pulse. We elected to use the chicken ASIC1 subunit for these experiments for two important reasons. First, cASIC1 is the same subunit used from structural studies (*Baconguis et al., 2014*; *Baconguis and Gouaux, 2012*; *Gonzales et al., 2009*; *Jasti et al., 2007*; *Yoder et al., 2018*). Second, in our hands, cASIC1 does not undergo the strong tachyphylaxis mammalian ASIC1a does in outside out patches (*Chen and Gründer, 2007*). Such strong tachyphylaxis prevents a thorough mapping of the recovery time course and non-stationary noise analysis (*see below*).

We initially examined the recovery time course of cASIC1 wild type and found that cASIC1 essentially completely desensitized with a time constant of $181 \pm 6$ ms (*Figure 1C,E–F*) and fully recovered in about 10 s ($\tau_{rec}$rec840 $\pm$ 90 ms, slope m = 0.96 $\pm$ 0.05, n = 5, *Figure 1C,E–F*). Consistent with our hypothesis that a smaller residue in the Leu414 position would be more nimble and subject to less steric hindrance, the L414A mutation underwent faster desensitization ($41 \pm 1$ ms, n = 5, $p<1e^{-5}$ vs wild type) but also recovered exceptionally fast. This can be seen in *Figure 1C* where an L414A patch is overlaid with a wild-type patch. At the shortest inter-pulse interval of 10 ms, wild-type channels show negligible recovery yet L414A has recovered by more than 50%. To properly resolve this highly accelerated time course, a modified pulse protocol was used with very short inter-pulse intervals (*Figure 1D*). This revealed L414A was essentially fully recovered in ~20 ms ($\tau_{rec}$rec4.0 $\pm$ 0.5 ms, m = 9 $\pm$ 3, n = 5, $p<1e^{-5}$ versus wild type, *Figure 1C–F*), or approximately 200 times faster than wild type. In past studies (*Kusama et al., 2013*; *Li et al., 2012*), recovery from desensitization has been well described as a mono-exponential process. This was the case for cASIC1 wild type;

however, the L414A mutation was poorly fit by a single exponential function (*Figure 1E*, dotted line), requiring the use of a Hodgkin-Huxley type fit with a slope greater than 1. The dramatic effect of L414A highlights the importance of the β11–12 linker in controlling both entry to and exit from the desensitized state.

Recently, *Wu et al. (2019)* reported that L414A in human ASIC1a slows or attenuates desensitization, as well as right shifts proton activation curves. Rightward shifts of activation curves have also been reported by Roy *Roy et al. (2013)* for Ala and other substitutions at the Leu414 position. These past results are somewhat surprising given that the Leu414 does not appreciably move between the open and resting structures (*Figure 1—figure supplement 1A*). Moreover, single or even combined mutations to the putative proton sensors in the thumb/finger and palm domain do not produce such robust shifts as those reported for the Leu414 position (*Liechti et al., 2010*; *MacLean and Jayaraman, 2017*; *Paukert et al., 2008*; *Vullo et al., 2017*). Therefore, we examined the proton sensitivity of cASIC1 wild type and L414A in outside out patches using fast piezo-driven perfusion (*Figure 1—figure supplement 1B–D*). Consistent with the lack of motion of Leu414 between resting and open states, but in contrast to previous reports, we observed only a small shift in the $pH_{50}$ of activation for L414A compared to wild type (wild type: $pH_{50act} = 6.57 \pm 0.03$, n = 5; L414A: $pH_{50act} = 6.43 \pm 0.03$, n = 6, p=0.027). However, we did observe that the desensitization of Leu414A is incomplete and a sustained current develops with pH values less than 5 (*Figure 1—figure supplement 1E–F*). This sustained current may be related to previous reports of non-selective currents which arise due to mutation (*Vullo et al., 2017*) or toxin/sulfhydryl treatment (*Baconguis and Gouaux, 2012*; *Gautschi et al., 2017*). Interestingly, the pH-dependence of this sustained current is comparable to that reported for L414A activation in oocytes (*Roy et al., 2013*; *Wu et al., 2019*). For example, at pH 4.4, the fold change in the steady state over peak current compared to pH 5 is almost doubled ($1.8 \pm 0.2$) and this rises to a six-fold increase at pH 4 ($6.1 \pm 1.4$) (*Figure 1—figure supplement 1E–F*). Therefore, past experiments with this, and likely other mutations with similar phenotypes, have probably reported the pH sensitivity of the emerging sustained current while the peak current would have desensitized too quickly to adequately resolve. The prominent phenotype of L414A revealed by fast solution exchange may also offer insight into the connectivity or preferred route for channels transitioning between the desensitized, open and protonated-closed states.

## Evidence that ASICs enter desensitization from closed states

ASIC gating has been broadly captured by a linear kinetic scheme where the desensitized state is connected solely to the open state (*Gründer and Pusch, 2015*; *MacLean and Jayaraman, 2017*). We considered whether ASICs might enter the desensitized state at low pH values by first passing through the open state, as implied by such a linear model, or desensitize from closed states as suggested by a branching model. We reasoned that if ASICs desensitize via open states, then the acceleration of entry into desensitization produced by L414A would shorten open lifetimes but may not appreciably reduce the peak open probability. That is, every channel would still have to open, and, provided that their open times are several milliseconds long, the peak open probability should not decrease substantially. However, if ASICs desensitize primarily from shut states, as suggested by a branching model, then accelerating desensitization should favor the desensitized branch at the expense of the open branch and thereby substantially reduce peak open probability. To test this idea, we turned to non-stationary fluctuation analysis (NSFA) to provide estimates of peak open probability and single channel conductance. An important requirement of NSFA is that the peak amplitude of the population response does not vary or run down excessively over several dozen sweeps. Tachyphylaxis of mammalian ASIC1a in patches precludes using NSFA. Fortunately, cASIC1 responses were very stable in our hands, allowing us to obtain four records of cASIC1 wild type of between 50 and 100 sweeps not varying in amplitude by more than 10% (see Materials and methods). *Figure 2A* illustrates one such patch where the variance for 50 consecutive sweeps was calculated and plotted as a function of the current amplitude (*Figure 2B*). The NSFA indicated a peak $P_{open}$ of $0.86 \pm 0.02$ (n = 5, *Figure 2C*) with an estimated single channel conductance of $10 \pm 1$ pS, consistent with previously published conductance data (*Lynagh et al., 2017*; *Zhang and Canessa, 2002*). Importantly, to our knowledge, this represents the first estimate of the open probability of recombinant ASICs. Consistent with our hypothesis, NSFA of L414A containing patches yielded a significantly reduced peak $P_{open}$ of $0.71 \pm 0.02$ (n = 5, $p<1e^{-5}$ versus wild type,

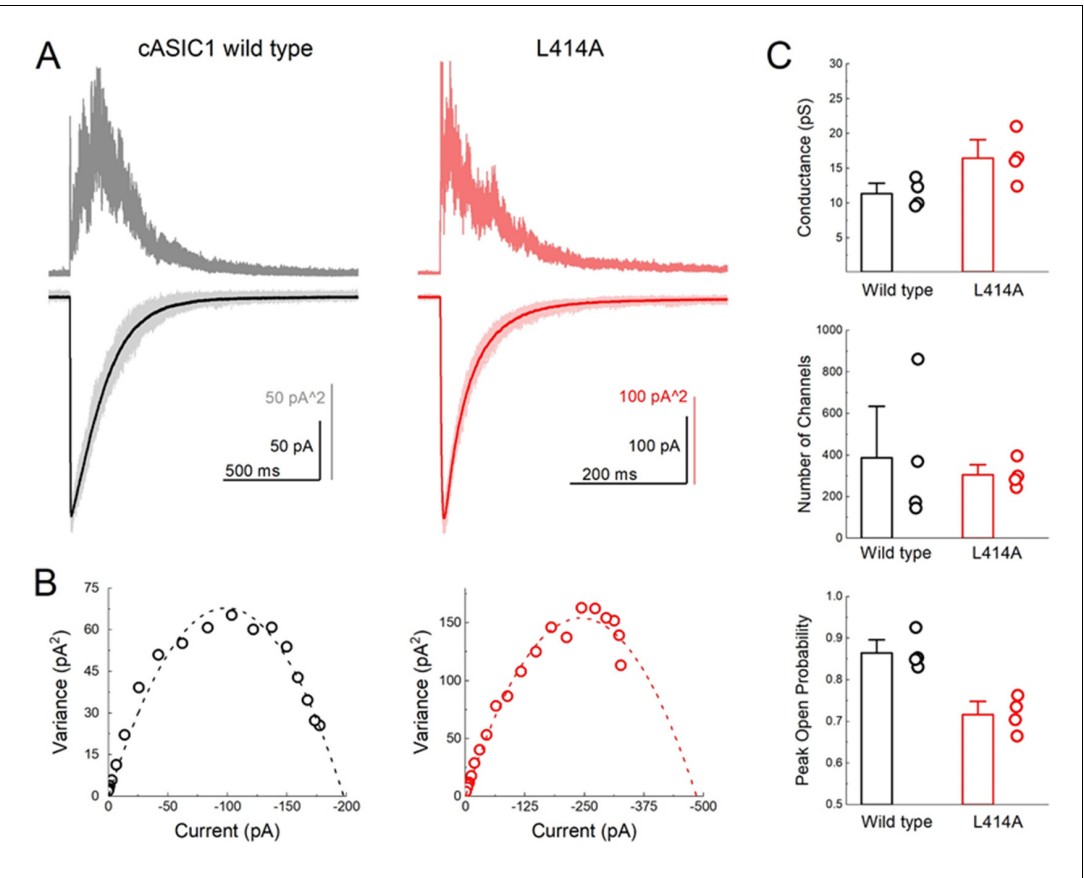

**Figure 2.** L414A lowers the open probability of cASIC1. (**A**) Representative recordings of wild type (*black and gray traces*) and L414A (*red traces*) during a non-stationary fluctuation protocol. The solid downward traces represent an average of approximately 70 individual sweeps, which are collectively shown in light color (*lower panels*). The upward traces are the ensemble variances from each patch (*upper panels*). (**B**) Current-variance plot from each patch in (**A**) with the dotted line depicting the fit to *Equation 4*. (**C**) Summary plots of conductance, the number of channels and the calculated peak open probability from each patch. The error bars are S.E.M. and the circles are individual patches.

*Figure 2*) with minimal change in single channel conductance (16 ± 2 pS, p=0.03 versus wild type). This result is consistent with ASICs desensitizing primarily from closed states. Thus, our experiments demonstrate that a smaller Ala residue at the Leu414 position imparts substantial effects on entry to and exit from desensitization, as well as changes in peak open probability without substantially altering proton potency (*Figures 1* and *2*, *Figure 1—figure supplement 1*). Taken together, these data argue that the 'flipping' motion of the β11–12 linker is crucial for desensitization but not activation. We next sought to explore the specific molecular interactions governing this flipping using molecular dynamics simulations.

## L414A destabilizes the β11–12 linker upon deprotonation

To test structurally whether the alanine mutation did in fact promote structural changes in the β11–12 linker region, we performed molecular dynamics simulations using the proposed structure of the desensitized state for wild type and with the L414A mutation. The desensitized state is expected to have a number of protonated acidic residues, however, the identity of these is unclear. We chose a protonation scheme which should stabilize the desensitized state (see Materials and methods) and compared this to the deprotonated state to ensure that the chosen protonation scheme stabilized the structure. From *Figure 3—figure supplement 1*, it is clear that the chosen protonation scheme (orange) stabilizes the structure better than the deprotonated state (blue), both when looking at the full protein structure and in particular when only including the extracellular domain. We therefore

simulated wild type and L414A channels in the desensitized state using this protonation scheme and examined the root mean squared deviation (RMSD) and fluctuation (RMSF) around the β11–12 linker region (*Figure 3A*). This analysis found a slight increase in the RMSD of the surrounding area induced by the L414A (*Figure 3A*, middle).

Next, we addressed whether L414A might exert some influence on linker stability under conditions mimicking recovery from desensitization. A challenge in this approach is that wild type, and even mutant L414A, ASICs recover in seconds or tens of milliseconds, well beyond the time frames amenable to molecular dynamics simulations. However, ASIC recovery from desensitization has been reported to depend on the pH separating the paired pulses (*Immke and McCleskey, 2003*; *MacLean and Jayaraman, 2016*). We observed a similar effect where the recovery from desensitization was slower when using pH 7.8 and 7.6 as the inter pulse pH ($\tau_{recovery}$ pH 7.8: 1600 ± 90 ms, m = 0.83 ± 0.01, n = 5, p=0.005 versus pH 8; pH 7.6: 11400 ± 600 ms, m = 0.73 ± 0.03, n = 5, p<1e$^{-5}$ versus pH 8, *Figure 3—figure supplement 2*). Interestingly, alkalinizing the interpulse pH was able to dramatically accelerate recovery in wild type channels. Specifically, when using pH 9, channels recovered with a time constant of 30 ± 2 ms (m = 3.0 ± 0.1, n = 5, p<1e$^{-5}$ versus pH 8), and this accelerated to 7.5 ± 0.4 ms (m = 5.3 ± 0.3, n = 5, p<1e$^{-5}$ versus pH 8) with pH 10 (*Figure 3—figure supplement 2*). Given this, we reasoned that if the simulated recovery conditions were sufficiently alkaline, then recovery from desensitization, or at least some initial phases of it, may be observable within the time frame of molecular dynamics simulations. Therefore, we simulated both wild type and L414A systems without protonating any acidic residues. In light of the promising results and relevance to our functional data, we extended these simulations relative to the protonated systems (*Figure 3B*). As each channel has three chains and we run three repeats of each setup, we in principle obtain data for nine chains (three chains x three repeats) for each setup. Upon

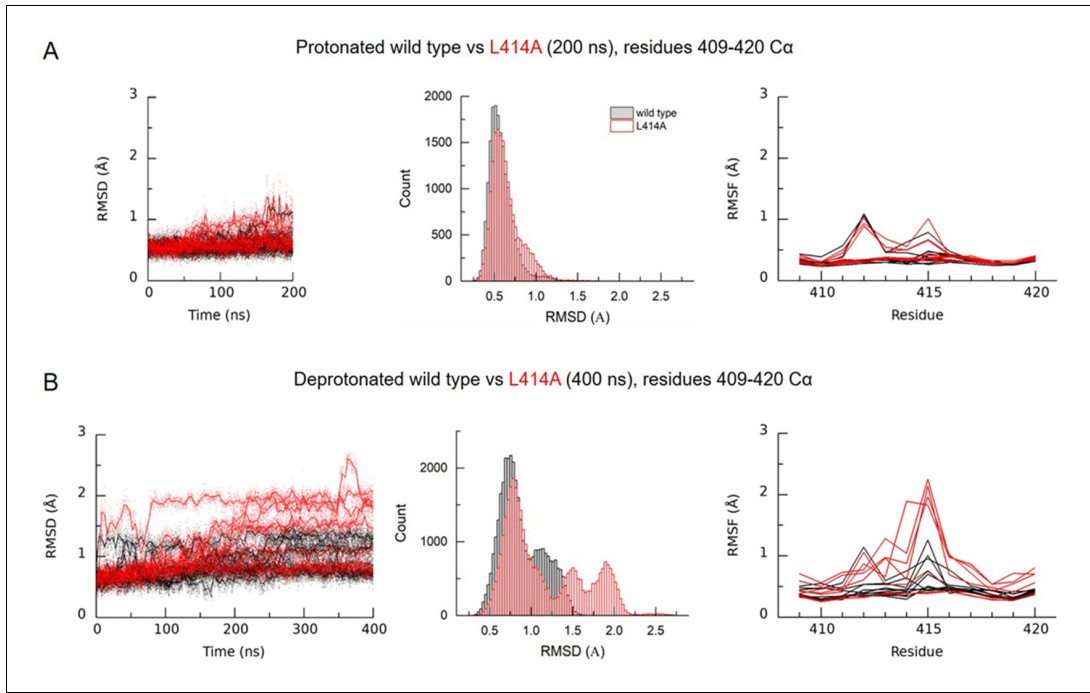

**Figure 3.** Deprotonation of L414A cASIC1 reduces the stability of the β11–12 linker in the desensitized state. (**A**) Calculated RMSD as a function of time (*left*) and across the entire simulation (*middle*) for C$_\alpha$ atoms of amino acid residues 409–420 of protonated cASIC1 wild type (*black*) or L414A (*red*) over 200 ns of simulation. (**A**, *right*) Calculated RMSF from the same simulations. Each chain from each of three independent runs is represented as a single trace. (**B**) Same as in **A**) but for deprotonated simulations, 400 ns.

The online version of this article includes the following figure supplement(s) for figure 3:

**Figure supplement 1.** The employed protonation scheme stabilizes the desensitized conformation of cASIC1.

**Figure supplement 2.** Recovery is accelerated by alkaline inter-pulse pH values.

**Figure supplement 3.** L414A recovery from desensitization has reduced pH-dependence.

deprotonation, we observed increased RMSD of the linker region for the wild type in three of the chains, while for the L414A mutant, six chains displayed increased RMSD values for the linker region relative to the protonated case. From RMSF calculations, it was also evident that the L414A mutation increased the flexibility of the middle part of the linker (*Figure 3B*). Furthermore, histograms of RMSD values at these residues revealed a sizeable destabilization induced by deprotonation in the L414A mutant compared to the wild type (*Figure 3B*). These molecular dynamics simulations support the interpretation that the alanine mutation accelerates recovery from desensitization by increasing the flexibility of the linker region.

The L414A mutation accelerated recovery from desensitization by approximately 200 fold (*Figure 1*). An alkaline conditioning pH of 10 also accelerated recovery from desensitization to a comparable extent in wild-type channels (*Figure 3—figure supplement 2*). Therefore, it is possible that some of the effect of L414A may arise because the mutation shifts the channel's pH-dependence of recovery. To test this, we examined the pH-dependence of recovery for L414A at pH 8, 7.8 and 7.6. We found that L414A retained very fast recovery at all pH's tested with a slight but significant slowing when using an inter-pulse pH of pH 7.6 ($\tau_{recovery}$ pH 7.8: $3.4 \pm 0.4$ ms, m = $10 \pm 2$, n = 6; pH 7.6: $8.4 \pm 0.6$ ms, m = $3.7 \pm 0.2$, n = 5, $p<1e^{-5}$ versus pH 8, *Figure 3—figure supplement 3*). Thus, pH-dependence of recovery is intact in this mutation but with a smaller effect over the examined range. This blunted pH-dependence of recovery in the L414A mutant channel suggests that L414A's acceleration of recovery is not an apparent effect arising from a shift in the pH-dependence but is a direct consequence of steric change at the 414 position.

## Hydrophobic patch stabilizes Leu414 position

In our simulations of the protonated desensitized state, we noted that the side chain of L414 interacted with the side chains of E80, A82, Q277, L281, Y283, I306, M364, V368 and R370 (*Figure 4—figure supplement 1*). These side chains form a cluster or pocket of residues with Y283 and R370 creating the 'back wall' toward the central axis of the channel, E80 and A82 forming the 'front wall' and Q277 contributing on one side (*Figure 4A*). The remaining 'side' of this patch is created by hydrophobic residues L281, I306, M364 and V368 and all interacting with the hydrophobic side chain of L414. We hypothesized that this 'hydrophobic patch' may stabilize the longer Leu side chain in the downward state but provide fewer interactions for the shorter Ala residue in the L414A mutant (*Figure 4—figure supplement 1*), possibly resulting in faster recovery. If this hypothesis is true, one expects that similar shortening of side chains on the hydrophobic patch side should also accelerate recovery from desensitization. To test this, we mutated each of the hydrophobic residues in that region to Ala and examined the recovery from desensitization. We found that all Ala mutations significantly accelerated recovery from desensitization, and most cases the effect was substantial. Specifically, the time constants (and slopes) for recovery from desensitization for L281A, I306A, M364A and V368A were $25 \pm 1.1$ (m = $3.9 \pm 0.2$, n = 5, $p<1e^{-5}$ versus wild type), $135 \pm 7$ (m = $1.31 \pm 0.04$, n = 5, $p<1e^{-5}$ versus wild type), $140 \pm 3$ (m = $1.44 \pm 0.02$, n = 4, $p<1e^{-5}$ versus wild type), and $520 \pm 50$ ms (m = $0.99 \pm 0.06$, n = 5, p=0.01 versus wild type), respectively (*Figure 4*). Interestingly, L281A showed the largest acceleration of recovery but also markedly increased the rate of channel desensitization ($\tau_{des} = 47 \pm 3$ ms, $p<1e^{-5}$ versus wild type). However, I306A and M364A did not substantially alter desensitization ($\tau_{des} = 240 \pm 11$; $108 \pm 3$ ms, respectively). We therefore made the double I306A/M364A mutation with the goal of dramatically altering recovery without effecting entry into desensitization. However, this double mutation did not exhibit an increased effect on recovery as compared to the single mutations ($\tau_{recovery}$ $110 \pm 9$ ms, m = $1.37 \pm 0.03$, n = 5, *Figure 4*). Nonetheless, these data suggest that the desensitized state is partially stabilized by interactions between the Leu414 in the downward position and an adjacent hydrophobic patch in the neighboring subunit. Furthermore, shortening the side chain from Leu to Ala may 'release' this residue from the hydrophobic patch, possibly accounting for increased interactions with nearby Q277 (*Figure 4—figure supplement 1*) and additional flexibility (*Figure 3*). We next set out to characterize the structure-activity relationship of amino acids at the 414 position itself.

## Side chain of 414 position impacts desensitization in complex way

The L414A substitution is a substantial reduction in size but also a small reduction in hydrophobicity of the 414 side chain. To systematically examine the impact of either of these dimensions, size and

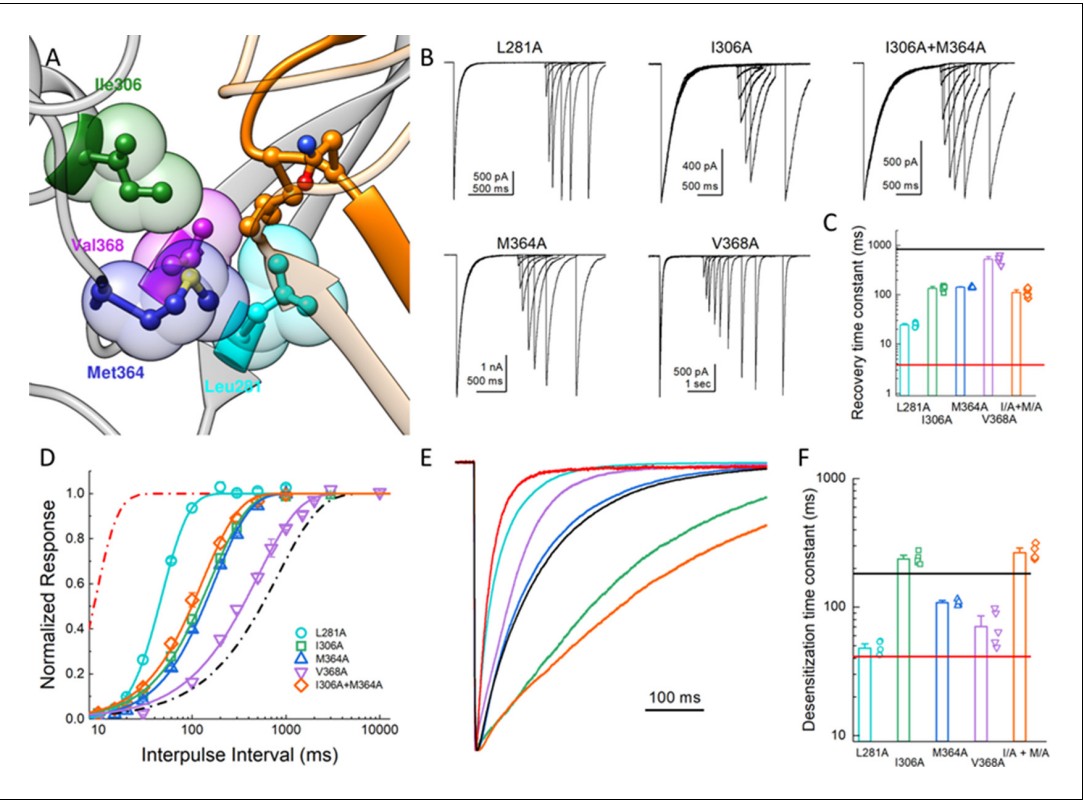

**Figure 4.** Hydrophobic patch influences the kinetics of desensitization and recovery. (**A**) Structure of the low-pH desensitized state (*PDB:4NYK*). The β11–12 linker is shown in orange. The adjacent subunit is depicted in gray with hydrophobic residues poised to interact with Leu414 shown in individual colors as ball and stick and partially transparent spheres. (**B**) Representative recordings of paired pulse protocols for Ala substitutions in the hydrophobic patch. (**C**) Summary of recovery time constants from all patches for each mutant. The black and red lines illustrate the time constants for wild type and L414A, respectively. (**D**) Summary of recovery curves as a function of inter pulse interval for each mutant. (**E**) Representative traces showing the entry into desensitization for each mutation as well as the time constants of entry for all patches (**F**). The solid lines show the time constants for wild type (*black*) and L414A (*red*). The error bars are S.E.M. and the circles are individual patches.

The online version of this article includes the following figure supplement(s) for figure 4:

**Figure supplement 1.** L414A reduces interactions with adjacent residues.

polarity, we mutated the 414 position to large charged (Arg), polar (Tyr), and hydrophobic (Phe) residues as well as a polar residue (Asn) to match the small non-polar Ala. We also substituted Leu414 for Ile, which has the same size with the same number of atoms and a similar hydrophobicity but differ in the branch point. If the only considerations at this position are size (*Wu et al., 2019*) and polarity, then one would predict that progressive increases in either dimension should slow entry and exit, yielding a rank order (fastest to slowest) of Ala, Leu = Ile, Asn, Phe, Tyr, Arg. We therefore repeated our recovery protocols anticipating this rank order. Instead, we found that no clear pattern emerged in either the entry to or exit from desensitization. L414R desensitized and recovered very slowly ($\tau_{des}$des1600 ± 380 ms; $\tau_{rec}$rec41000 ± 6400 ms, m = 0.9 ± 0.1, n = 3, p<1e$^{-5}$ versus wild type for both, *Figure 5*) as predicted from the side chain and as expected given the single channel open durations (*Wu et al., 2019*). However, every other mutation ran counter to the simple hypothesis that size and polarity alone predict desensitization (*Figure 5*). In contrast to expectations and previous reports using two-electrode recordings in oocytes (*Roy et al., 2013*; *Wu et al., 2019*), the large aromatic side chain substitutions of Phe and Tyr actually resulted in much faster entry and exit in both cases (L414F: $\tau_{des}$des5.6 ± 0.2 ms; $\tau_{rec}$rec21 ± 2 ms, m = 3.4 ± 0.6, n = 5; L414Y: $\tau_{des}$des3.2 ± 0.2 ms; $\tau_{rec}$rec29 ± 2 ms, m = 2.7 ± 0.2, n = 4, p<1e$^{-5}$ for all comparisons to wild type, *Figure 5*). Similarly, L414N was expected to enter and exit slightly slower than Ala; however, it

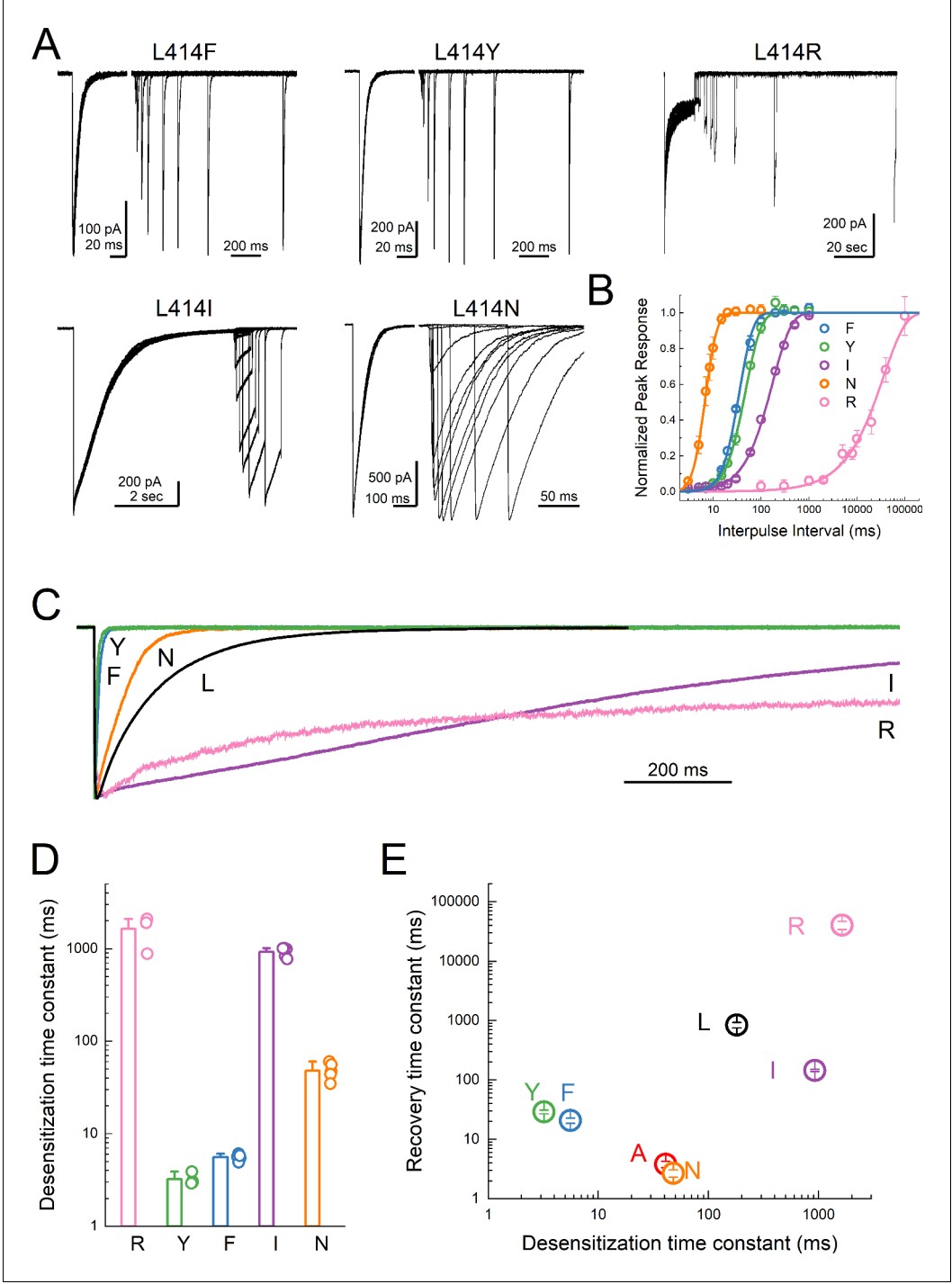

**Figure 5.** Substitutions at the Leu414 position produce a wide range of desensitization kinetics. (**A**) Representative recordings of recovery from desensitization protocols for a range of Leu414 substitutions. Note that for L414F, L414Y and L414N, the x axis has been broken into two different time scales (**B**) Summary of recovery as a function of inter pulse interval for all mutations. (**C**) Example traces of desensitization kinetics for the mutant panel. (**D**) Summary of desensitization time constants across all patches. The error bars are S.E.M. and the circles are individual patches. (**E**) Log-log scatter plot of the desensitization time constant and the recovery time constant for each mutation. Note that large side chains, ie. R, F or Y, can produce either very fast or very slow kinetics. The online version of this article includes the following figure supplement(s) for figure 5:

**Figure supplement 1.** Simple kinetic model cannot recapitulate ASIC gating.

showed comparable behavior ($\tau_{des}$des48 ± 3 ms; $\tau_{rec}$rec2.7 ± 0.4 ms, m = 11 ± 4, n = 6). The L414I substitution was an additional surprise. If the only factors at play are size and polarity, then this mutation should have minimal effect. However, we found that the L414I construct entered desensitization 5-fold slower ($\tau_{des}$des920 ± 50 ms, p<1e$^{-5}$ versus wild type) and recovered nearly 6-fold *faster* ($\tau_{rec}$rec145 ± 6 ms, m = 1.4 ± 0.04, n = 5, p<1e$^{-5}$ versus wild type) than wild type. Based on this surprising set of results, particularly the dramatic acceleration by the bulky Tyr residue and the notable effect of the conservative Ile substitution, we conclude that no simple rule of size or polarity is sufficient to explain or predict the effects of this position as yet. These results also stand in contrast to those reported recently and to the expectations of the purely steric 'valve' model (*Wu et al., 2019*). An important tenant of this model is that the swivel of the β11–12 linker occurs via an inward path toward the central axis of the pore. To examine this issue, we further analyzed our previous molecular dynamics simulations.

## Molecular dynamics simulations suggest Leu414 and N415 primarily transits in an 'outward' path

The RMSD analyses in *Figure 3* suggested that some chains, especially in the deprotonated states, underwent larger structural changes in the linker region. We investigated visually whether a flip of the L/A414 and N415 residues was observed in any of these chains. Given that each setup was repeated three times and that each protein has three chains, nine chains could be studied for each of the four setups. In the simulations of the wild type and the L414A mutant in protonated states, elevated RMSD values were observed for one chain in the wild—type simulations and three chains in the simulations of the mutant (*Figure 3A*). However, none of these showed any particular displacement of the L/A414 and N415 residues. For the simulations of the deprotonated wildtype protein, elevated RMSD values were observed for three chains (*Figure 3B*). Two of these showed no signs of flips, while for the third chain, L414 and N415 seemed to come close to flipping, following the outwards path as originally suggested by *Yoder et al. (2018)*; however, the conformation reverted back to the desensitized conformation before completing the flip. This is also evident when plotting the RMSD to the resting state against RMSD to the desensitized state (*Figure 6*; *Figure 6—figure supplement 1*). Finally, in agreement with L414A increasing flexibility of the β11–12 linker, elevated RMSD values for the linker region are observed for six of the nine chains for the deprotonated L414A mutant. One of these chains showed no signs of flipping, while the other five chains underwent at least a 'partial flip' (*Figure 6—figure supplement 1*). In three of these chains, a partial flip was observed as N415 flipped over without A414 undergoing the full flip. In all of these cases, N415 followed an outwards path. In one chain, a relatively full flip was observed with N415 taking an inwards path and A414 an outwards path. Finally, the full flip was observed for one chain in which a very good overlay with the resting conformation of the linker was obtained (*Video 1*). In this trajectory, the outwards path was followed for both residues. Hence, while we would need more repeats and more full flips to get conclusive data, our work supports the original suggestion of an outwards path for both L414 and N415.

## ncAA incorporation and UV-trapping at Leu414

Finally, we addressed whether the flipping of the β11–12 linker is the sole means of channel desensitization or if some other mechanisms may also participate. Previous work has attempted to use disulfide trapping of Leu414 and the adjacent Leu86 to investigate the requirement of flipping for desensitization (*Yoder et al., 2018*). However, putative disulfide trapping between these two residues resulted in partial suppression of desensitization, possibly indicating other mechanisms are at play. Therefore, to investigate the necessity of β11–12 flipping in desensitization, we turned to noncanonical amino acid (ncAA) incorporation and UV trapping. The ncAA *p*-benzoyl-L-phenylalanine (Bpa) generates a free radical when exposed to 365 nm light (*Klippenstein et al., 2014*; *Pless and Ahern, 2013*; *Ye et al., 2008*). The resulting free radical spontaneously forms a covalent bond with a nearby atom, preferentially reacting with C-H bonds. Incorporation and trapping by Bpa has been previously used to investigate conformational changes in AMPA receptors (*Klippenstein et al., 2014*; *Poulsen et al., 2019*) and K$^+$ channels (*Murray et al., 2016*; *Westhoff et al., 2017*). If the 'flipping' of the linker is the sole determinant or mechanism of channel desensitization, then in principle one should be able to minimize macroscopic desensitization by UV-mediated trapping during

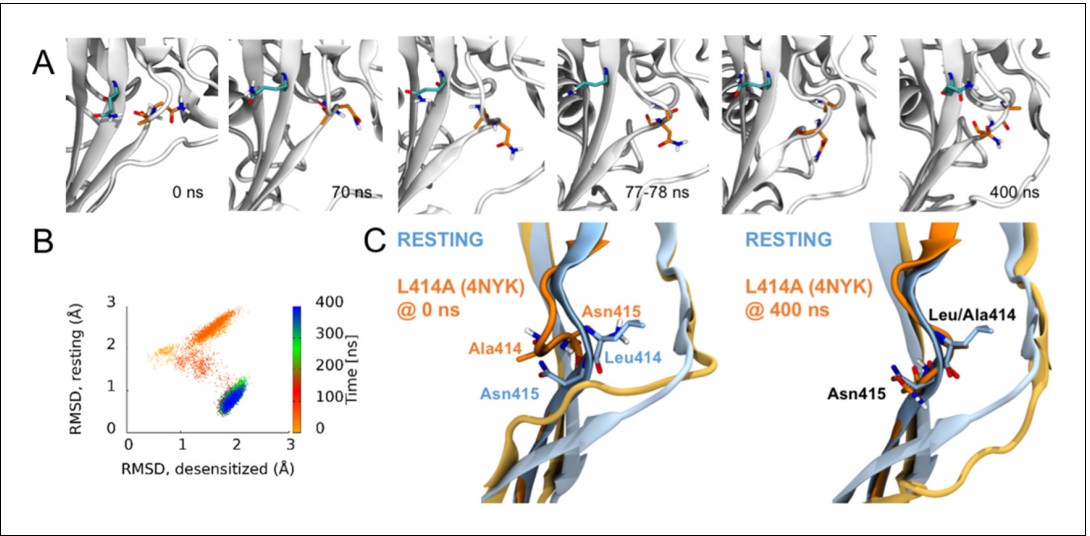

**Figure 6.** Side chains of 414 and 415 tend to flip along an outward path upon deprotonation of acidic residues. (**A**) Snapshots of chain C from repeat c at the indicated time points. A414 and N415 are shown in orange while Q277 is drawn in teal. (**B**) Calculated Cα RMSD values for amino acid residues 409–420 compared to the desensitized state (*x axis*) and resting state (*y axis*) for chain C from repeat c. The dots are colored according to simulation time as illustrated on the right-hand color bar. (**D**) Structural alignments of initial (*left*) and final (*right*) chain C from repeat c positions (*orange*) compared to resting (*blue*) state.

The online version of this article includes the following figure supplement(s) for figure 6:

**Figure supplement 1.** Transitions from desensitized to resting-like states upon deprotonation.

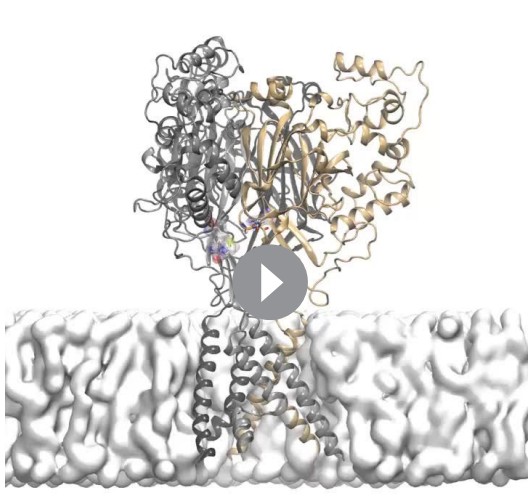

**Video 1.** Animation illustrating the β11–12 linker flip. Only the simulation period 50–90 ns is included. A414 and N415 are shown in licorice with orange carbon atoms. The hydrophobic patch (Leu281, Ile306, M364 and V368) on the neighboring chain, as well as Q277 on the same chain (behind A414 and N415), are illustrated in licorice with gray carbon atoms and transparent surface representation.

https://elifesciences.org/articles/51111#video1

the resting state pH 8 application. This should result in locking the 414 position, and hence the β11–12 linker, in an upward position and preclude desensitization. To test this, we first attached a GFP to the C terminus of wild-type cASIC1 (cASIC_GFP) and subsequently mutated Leu414 to contain the amber stop 'TAG' codon. Since the C terminal GFP should only fluoresce once the upstream channel has been translated, the GFP emission should theoretically correlate with 'stop' suppression and channel rescue. For 'stop' suppression and ncAA incorporation, we combined two copies of cDNA encoding the Bpa tRNA and a single copy of the cDNA encoding Bpa synthetase into a pcDNA 3.1 vector. This vector, termed R3, provided a single package for delivering all Bpa incorporation machinery into mammalian cells. This vector, as well as either wild type or L414TAG cASIC_GFP plus an additional Bpa tRNA construct, YAM, was transfected into cells with or without a methyl ester variant of Bpa (MeO-Bpa) added to the culture media. As seen in *Figure 7—figure supplement 1*, cASIC1_GFP showed robust fluorescence emission which exhibited localization consistent with plasma membrane distribution. Transfection of cASIC1_GFP L414TAG with R3 but not MeO-Bpa or with MeO-Bpa but not R3 produced

detectable GFP fluorescence but the signal was diffuse, consistent with a soluble protein and not a membrane protein (*Figure 7—figure supplement 1*). Our interpretation of this is that one of the four Met residues after the 414 position acts as an alternative start codon, allowing GFP translation. However, combining the template, tRNA, synthetase and MeO-Bpa recovered the GFP fluorescence and localization consistent with plasma membrane distribution (*Figure 7—figure supplement 1*).

In outside-out patch experiments, GFP-positive cells transfected with template, synthetase and tRNA as well as with MeO-Bpa added yielded very small currents which were difficult to resolve in a convincing fashion. We therefore turned to whole cell recordings to increase the measured responses. In whole cell configuration, these cells gave rise to resolvable currents that exhibited rapid activation and near complete desensitization (*Figure 7A and C*). To test for UV-induced modulation, we first applied a pH step from pH 8 to 5 for five successive jumps to get a stable baseline. Subsequently, 14 pulses of 50 ms duration of UV LED were applied to the cell over 7 s prior to the agonist application, allowing for the large majority of channels to be in the resting state during exposure (i.e. resting state, see Materials and methods). As seen in *Figure 7*, such UV application produced a strong and immediate slowing of desensitization and robustly reduced the extent of desensitization (*Iss/Ipeak*, n = 7, p<$1e^{-5}$ between pre and post-UV), both of which would be expected if the β11–12 linker flipping was a requirement for channel desensitization. Cells transfected with cASIC1_GFP L414TAG plus R3 but without MeO-Bpa did respond to pH application (mean peak current −180 ± 70 pA, n = 4 compared to −340 ± 80 pA, n = 7, with MeO-Bpa). Crucially, these cells did not exhibit UV modulation (n = 4, p=0.7 between pre and post UV) nor did responses from cells expressing wild-type cASIC_GFP (n = 3, p=0.6 between pre and post UV), indicating the UV effect was specific to the incorporated Bpa (*Figure 7B and D*). These data indicate

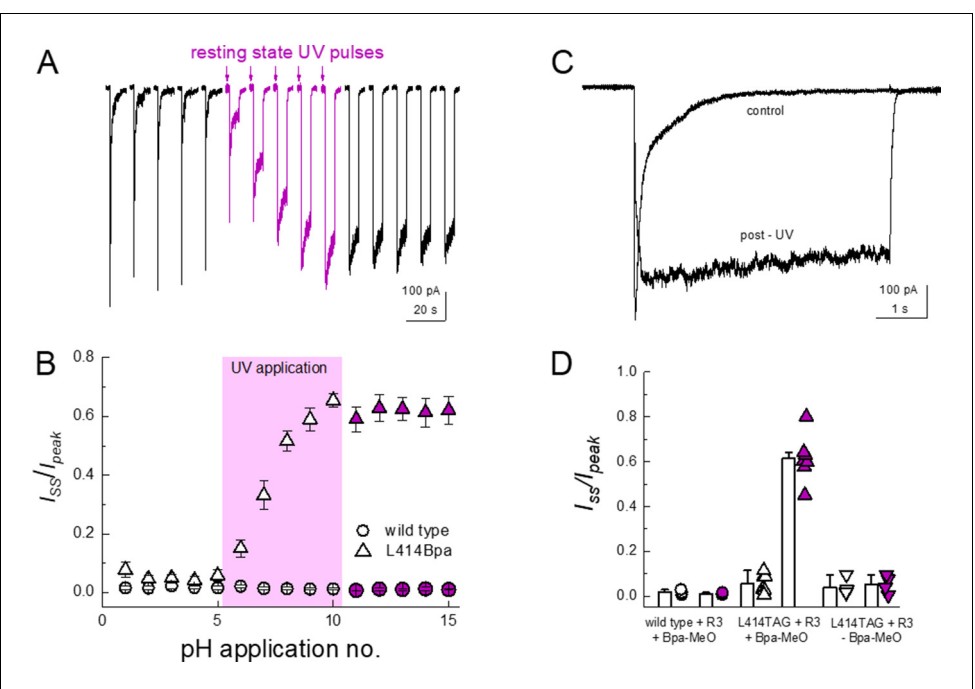

**Figure 7.** Resting state UV application suppresses the desensitization of L414Bpa. (**A**) Representative whole cell recording of cells transfected with L414TAG plus R3 and YAM and supplemented with MeO-Bpa and responding to pH 5 application. Following five pH 5 applications, high-power UV light is pulsed while the channels are in the resting state for additional pH 5 applications (*purple traces*) followed by applications without UV. (**B**) Summary of steady state current divided by peak current during pH 5 application before, during and after UV for L414Bpa (*triangles*) or wild type (*circles*). (**C**) Example responses from the same cell as **A**), before and after UV application. (**D**) Summary of steady state current divided by peak during pH 5 application for wild-type patches with MeO-Bpa (*circles*), L414TAG with MeO-Bpa (*upward triangles*), or L414TAG without MeO-Bpa (*downward triangles*).
The online version of this article includes the following figure supplement(s) for figure 7:

**Figure supplement 1.** Schematic of non-canonical amino acid incorporation.

that trapping the conformation of the 414 position in the resting state substantially reduces the capacity of the channel to undergo desensitization but does not preclude activation. Interestingly, the current rise times following UV application did change (10–90% rise time: before UV, 12 ± 1 ms; post UV, 830 ± 400 ms, n = 7, p<1e$^{-5}$ between pre- and post-UV, *Figure 7C*), indicating some rearrangements near the β11–12 linker accompany channel activation.

We next tested the possibility of UV trapping in the desensitized state by applying a similar UV pulse protocol at the end of a pH 5 application, when the channels had essentially completely desensitized. We found that desensitized state trapping required a stronger UV stimulus (more sweeps with UV and more UV pulses per sweep, see Materials and methods) but we did observe a strong inhibition of the peak response through the course of the UV application. As seen in *Figure 8*, UV application to desensitized channels inhibited peak responses by approximately 50% (47 ± 4% peak response, n = 6, *Figure 8*) which was not observed in recordings from either wild-type cASIC1_GFP transfected cells (96 ± 3% peak response, n = 3) or in cASIC1_GFP L414TAG transfected cells when MeO-Bpa was omitted (91 ± 8% peak response, n = 3, *Figure 8D*). The residual current likely reflects a combination of incomplete trapping or endogenous ASIC currents. Combined with the results of the resting state trapping experiments (*Figure 7*), these data are clear evidence that the Leu414 position undergoes a sizeable motion between resting/open and desensitized states. Furthermore, the sizeable attenuation of macroscopic desensitization by resting state trapping experiments (*Figure 7*) argues that this β11–12 linker is the sole mechanism for channel desensitization.

## Discussion

In the present study, we investigated the molecular underpinnings of entry to and exit from desensitization in cASIC1. We corroborate and extend structural and functional studies implicating the β11–

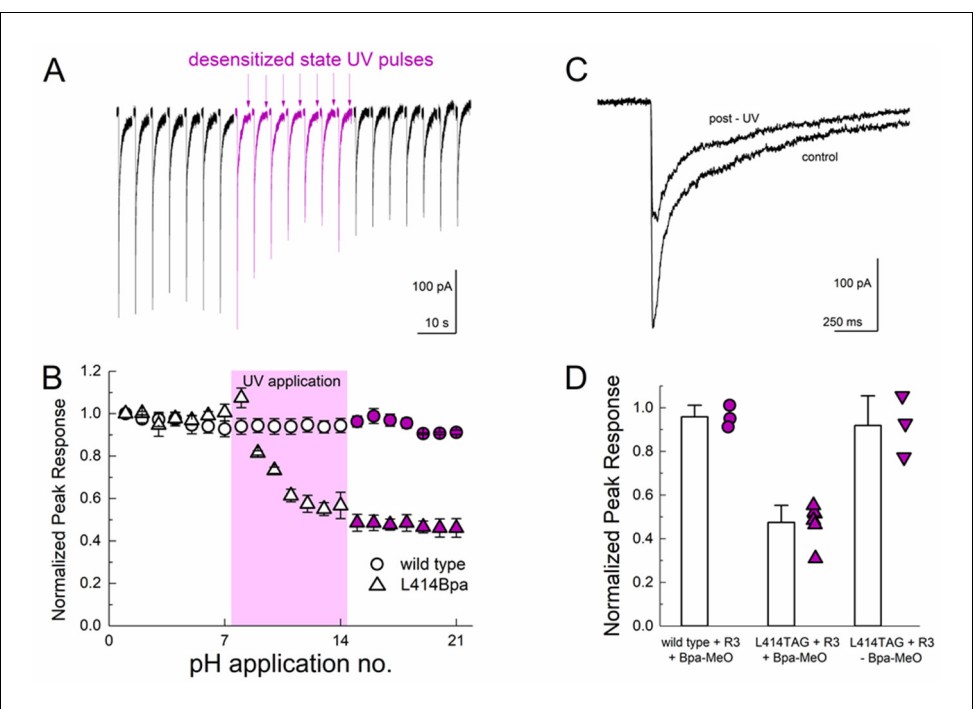

**Figure 8.** Desensitized state UV application inhibits L414Bpa peak responses. (A) Representative whole cell recording of cells transfected with L414TAG plus R3 and YAM and supplemented with MeO-Bpa and responding to pH 5 application. At the end of the eighth pH 5 application, UV light is pulsed while the channels are in the desensitized state. UV trains are applied for seven consecutive pH 5 applications (*purple traces*) followed by applications without UV. (B) Summary of normalized peak currents evoked during pH 5 application before, during and after UV for L414Bpa (*triangles*) or wild type (*circles*). (C) Example responses from the same cell as A), before and after UV application. (D) Summary of normalized peak responses for wild type cells with MeO-Bpa (*circles*), L414TAG with MeO-Bpa (*upward triangles*), or L414TAG without MeO-Bpa (*downward triangles*).

12 linker as a regulator of desensitization. Indeed, we report that a simple L414A mutation imparts a 5-fold and 200-fold acceleration in entry into and exit from desensitization, respectively (*Figure 1*). The acceleration of desensitization was strong enough to curtail peak open probability (*Figure 2*). We further highlight how this mutation does not appreciably affect the pH dependence of activation (*Figure 1—figure supplement 1*) but does destabilize the adjacent region upon deprotonation in the desensitized state (*Figure 3*). This destabilization appears to be sufficient to drive a conformational 'flip' of the 414 residue along with its neighbor, N415, to a conformation resembling the resting state (*Figure 6*). Furthermore, we identify a patch of hydrophobic residues in the adjacent subunit which impact the stability of Leu414 in the downward/desensitized conformation (*Figure 4—figure supplement 1*) and also the macroscopic rates of channel desensitization and recovery (*Figure 4*). We also demonstrate that structural alterations to the 414 position itself produce a range of kinetic effects, with no discernable pattern based on size or polarity (*Figure 5*). Finally, we employ state-dependent ncAA trapping to provide evidence that β11–12 flipping is the sole mechanism for channel desensitization (*Figures 7* and *8*).

## Comparison with previous studies

The β1–2 and β11–12 linkers are important determinants of ASIC gating (*Bonifacio et al., 2014*; *Coric et al., 2003*; *Gwiazda et al., 2015*; *Li et al., 2010a*; *Li et al., 2010b*; *Roy et al., 2013*; *Springauf et al., 2011*; *Wu et al., 2019*). In particular, the critical role of the β11–12 linker was first proposed by Baconguis et al., based on the observed swivel of the L414 and N415 positions between open and desensitized structures. This was further supported by mutations to either L414 or N415, which alter desensitization kinetics and/or extent (*Li et al., 2010a*; *Roy et al., 2013*; *Wu et al., 2019*). It has also been suggested that the β11–12 linker is an important determinant of activation as certain mutations (L414F, Y and A for example) have been reported to profoundly shifted activation curves measured in oocytes (*Roy et al., 2013*; *Wu et al., 2019*). We suggest that such large apparent shifts in activation curves arise due to the slower solution exchange of oocytes, the extremely rapid desensitization of such mutations (*Figures 1* and *5*) and the emergence of a sustained current with a right shifted pH-dependence (*Figure 1—figure supplement 1*). It is also possible that species-specific mutational effects contribute to these different phenotypes. However, we did observe that the activation times of these 'fast' mutants were generally faster than wild type (10–90% rise time: wt, 7 ± 3 ms, n = 15; L414A, 4 ± 1 ms, n = 11; L414Y 1.0 ± 0.3, n = 4). Further, the rise times of whole cell L414Bpa responses were much slower following resting state UV trapping than before (10–90% rise time: before UV, 12 ± 1 ms; post UV, 830 ± 400 ms, n = 7, p<1e$^{-5}$ between pre- and post-UV) and this effect was not observed with cASIC1_GFP (10–90% rise time: before UV, 10 ± 6 ms; post UV, 10 ± 6 ms, n = 3). Finally, the slope of the L414A pH activation curve is shallower than that of wild-type cASIC1 (*Figure 1—figure supplement 1C*). Thus, there are likely some local re-arrangements of the β11–12 loop during the resting to open transition which Leu414 can regulate; however, these re-arrangements comprise a smaller portion of the energy barrier than was previously proposed.

It has recently been suggested that the mechanism of ASIC desensitization can be likened to a valve mechanism, wherein Gln277 acts as a clamp or valve controlling the β11–12 linker flip, and hence desensitization (*Wu et al., 2019*). In this model, upon protonation/channel activation Gln277 moves slightly away from Asn415, which allows the swivel of β11–12 linker and desensitization to occur. Once in the desensitized state, the Q277 'valve' shuts to prevent β11–12 reverting to the resting/open state conformation and channel re-activation. This accounts for the essentially complete desensitization of the channel. This model suggests that desensitization is purely determined by steric forces between Gln277, Leu414 and Asn415 and requires that Leu414 and Asn415 both swivel inwards, towards the central axis of the channel, as opposed to the outward motion proposed by Yoder et al. The phenotype of Q277G uncovered by Wu et al. is quite striking and will undoubtedly be useful to the field; however, the 'valve' model is inconsistent with several observations. First, if the prime determinant of desensitization is the size of the 277 side chain, then one would expect Gln277 to behave identically to Glu277, and Asn277 to be identical to Asp277. However, mutations bearing the acidic side chains show slower/reduced extent of desensitization than their amide counter parts, hinting at a role for electrostatic interactions (*Wu et al., 2019*). Second, if the transition from the open to desensitized state (or protonated-closed to desensitized state) requires a β11–12 linker flip inwards and not outwards, one would expect to see some sort of cavity or pathway on

the inward route and not the outward face to permit this flip. However, examining the calculated surfaces of the open (or resting) and desensitized states, a clear transit path is observed along the outward trajectory proposed by Yoder et al. but not the inward route posited by Wu et al. This makes the inward path a less probable route, although we cannot exclude some short-lived intermediate state where an inward path opens. Third, in our molecular dynamic simulations repeats where β11–12 flipping is observed, we generally see a clear outward transit for the 414 and 415 side chains (*Video 1*). Fourth and finally, if Gln277 acts as a valve to prevent β11–12 flipping, one might expect its interaction with the linker to be quite stable in the desensitized state. However, our simulations with the wild type and L414A protonated states suggest that this might not be the case. While the side chain conformation of Gln277 is relatively stable in the simulations with protonated states, as judged from the $\chi_1$- $\chi_3$ side chain dihedral angles (*Figure 9*), a number of changes in the side chain conformations are observed. Such conformational flexibility of Gln277 occasionally causes its side chain to move outside of hydrogen bond distance between Gln277Nε and Leu414O (*Figure 9*). An alternative hypothesis is that Gln277 acts as a hydrogen bond partner to stabilize the β11–12 linker in the desensitized conformation, and not as a purely steric valve. However, this hydrogen bond hypothesis would predict only a smaller effect of Q277G, not a complete abolishment of desensitization that has been reported (*Wu et al., 2019*). Clearly, as revealed by our data and previous reports, β11–12 linker flipping is an integral component, indeed perhaps the sole mechanism, of channel desensitization. Future pairings of molecular dynamic simulations and kinetic experiments may shed light on how the precise interplay between Gln277 and the β11–12 linker shapes the desensitization process.

## Does ASIC desensitization proceed from open or shut states?

Rapid application of acidic solution activates ASICs, while also leading to their desensitization. However, it is unknown if this low pH desensitization proceeds from the open state or from a protonated-closed state, nor is it known if the channels recover by passing through the open state. ASIC gating has been broadly captured by a linear kinetic scheme where the desensitized state is connected solely to the open state (*Gründer and Pusch, 2015*; *MacLean and Jayaraman, 2017*). This model predicts that as channels recover, they are forced to transiently pass through the open state. Presumably, these openings would be spread across the 10 s recovery period in a wild-type channel

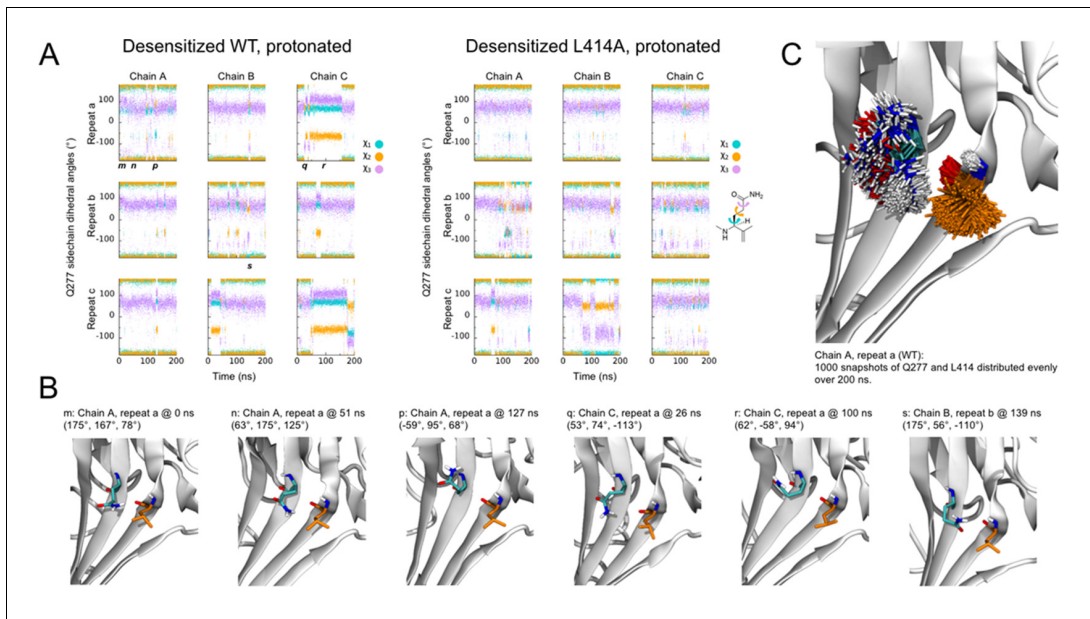

**Figure 9.** Flexibility of Gln277 residue. (**A**) Dihedral angles for the Gln277 side chain from wild type (*left*) and L414A (*right*) simulations under protonated (i.e. desensitized) conditions. (**B**) Snapshots at the indicated time points from wild-type simulations to illustrate various side chain configurations. (**C**) Overlay of side chain positions from the indicated simulation. Gln277 is shown in teal, Leu414 in orange.

and hence easy to miss. However, if this scheme is correct, then the rapid recovery of L414A would require the entire channel population to pass back through the open state within the 20 ms recovery period, giving rise to either a dramatic slowing of deactivation or a resurgent current following agonist removal. We observed neither of these phenomena in L414A (*Figure 1C,D*) or any other fast recovering mutant in our study, suggesting that ASICs more likely recover from low pH desensitization by passing through closed states and not open states. In addition, we also observed that the accelerated desensitization of L414A curtails peak open probability (*Figure 2*). As noted above, this finding is more consistent with desensitization stemming from closed and not open states. However, additional experiments, ideally at the single channel level, are needed to support this hypothesis. Furthermore, a simple closed-state desensitization branching mechanism also fails to recapitulate other aspects of our data.

## The recovery from desensitization process

Kinetic models generally predict that increasing the rate constant for recovery tends to increase the amplitude of the steady-state current (*Figure 5—figure supplement 1A,B*; *Carbone and Plested, 2012*; *Goldschen-Ohm et al., 2010*; *Robert and Howe, 2003*). Yet some of our mutations accelerate recovery by more than two log units with no substantial increase in steady-state current. Why not? A possible explanation maybe because the recovery process is itself pH-dependent. In general, steady-state currents arise as agonist-bound channels escape desensitization and re-open. However, in ASICs the escape process, recovery from desensitization, becomes progressively slower with increasing acidity (*Figure 3—figure supplement 2*). If this trend is extrapolated from pH 7.6 to pH 7, then on to pH 6 and pH 5, then the route out of desensitization may become progressively less and less favorable, giving rise to a near complete desensitization at pH 5 yet rapid recovery at pH 8. We propose that the pH-dependence of recovery therefore underlies the minimal steady-state current observed in mutations with extremely fast exit kinetics.

A second observation which emerges from our data is that L414A is poorly fit by a single exponential and requires a very steep slope (*Figure 1*). In fact, we found that for all our mutations and conditions 'faster' recoveries were best fit by rapid time constants and steeper slopes, the magnitude of which were correlated (*Figure 5—figure supplement 1C*). In contrast, simply accelerating the microscopic rate constant for recovery in a four-state model produced faster time constants with no substantial change in slope (*Figure 5—figure supplement 1C*). While the mechanistic basis for this relationship is unclear, it indicates that ASIC recovery from desensitization cannot readily be explained by a simple one-step process under all conditions. Rather, multiple steps and some degree of cooperativity, either at the level of protonation or subunit transition, are deeply embedded in the ASIC recovery process even if not always apparent. These observations, along with past reports, highlight the complexity and sensitivity of ASIC gating. It remains to be seen which, if any, of these highly pH-sensitive kinetic features serve a physiological role.

## Conclusion

Using a combination of fast perfusion electrophysiology, mutagenesis and molecular dynamics simulations, we highlight the crucial role of Leu414, and the surrounding area, in the kinetics of ASIC desensitization and recovery. We also employ ncAA-based photo-modulation to drive the state-dependent trapping of Leu414 into either the resting/open states, thus minimizing desensitization (*Figure 7*) or into the desensitized conformation, reducing channel activation (*Figure 8*). Based on our and previous work, it is clear that the β11–12 linker and surrounding areas act as a switch or clutch (*Yoder et al., 2018*) to control the desensitization process of ASICs. While our data provides support for an outwards transit path for L414 and N415 when undergoing the suggested conformational 'flip', many of the precise molecular details remain unclear. For example, why do large benzene ring substitutions (i.e. L414F) display fast kinetics comparable to small amino acid side chains? In addition, despite clear indication that deprotonating the desensitized state destabilizes the 'down/desensitized' conformation of L/A414, the upstream molecular forces which drive the β11–12 linker to become more stable in the 'down/desensitized' versus the 'up/resting' configuration during desensitization remain to be determined. What are the precise sets of protonation states and residues which govern this equilibrium? What are the relative contributions of the acidic pocket, the palm domain, the anion binding site and the β-ball in driving linker flipping? Future work combining

molecular dynamics simulations, fast perfusion electrophysiology and state-dependent modulation may yield further insight into these issues.

# Materials and methods

## Key resources table

| Reagent type (species) or resource | Designation | Source or reference | Identifiers | Additional information |
|---|---|---|---|---|
| Cell line | HEK293T | ATCC | CRL-3216 | |
| Recombinant DNA reagent | pcDNA3.1+-cASIC1 | Gifted from Dr. Vasanthi Jayaraman PMID: 24196950 | | |
| Recombinant DNA reagent | pcDNA3.1+-codon-optimized-cASIC1_GFP | This paper | | cASIC1 codon-optimized for mammalian expression tagged with C-terminal GFP |
| Recombinant DNA reagent | Codon-optimized-cASIC1_GFP | Integrated DNA Technologies | | Linear fragment used in assembly of pcDNA3.1+-codon-optimized-cASIC1_GFP via NEBuilder |
| Recombinant DNA reagent | R3 plasmid | Custom gene synthesis | | Contains 2 copies of Bpa tRNA and one Bpa tRNA synthetase (BpaRS) based on PMID: 17993461 |
| Recombinant DNA reagent | YAM Bpa tRNA | Gifted from Dr. Vasanthi Jayaraman | | Contains 1 copy of Bpa tRNA |
| Commercial assay or kit | Q5 Hot Start High-Fidelity 2X Master Mix | New England Biolabs, inc | M0494L | PCR |
| Commercial assay or kit | KLD Enzyme Mix | New England Biolabs, inc | M0554S | Mutagenesis ligation |
| Commercial assay or kit | NEBuilder HiFi DNA Assembly Master mix | New England Biolabs, inc | E2621L | Assembly of codon-optimized cASIC1_GFP |
| Chemical compound, drug | Polyethylenimine 25 k | Polysciences, Inc | 23966–1 | Transfection reagent |
| Chemical compound, drug | jetPRIME | Polyplus Transfections | 114–15 | Transfection reagent |
| Chemical compound, drug | Benzylphenylalanine methyl ester (MeO-Bpa) | Parent Bpa purchased from Bachem, conjugated to methyl ester by Dr. Chris Ahern | | Noncanonical amino acid methyl ester derivative (40 µM) |
| Sequence-based reagent | cASIC_P250 | This paper | Sequencing primer | CCACAGCCAGGATCCTCCACTCATCG |
| Sequence-based reagent | Codon-optimized cASIC_P350 | This paper | Sequencing primer | CATTTCTTGGTTGAAAAGG |
| Software, algorithm | PyMOL | | RRID:SCR_000305 | Structure modelling |
| Software, algorithm | UCSF Chimera | | RRID:SCR_004097 | Structure modelling, figures |
| Software, algorithm | GROMACS v 5.0.7 | http://www.gromacs.org/Downloads | RRID:SCR_014565 | Simulations |
| Software, algorithm | Axograph | | RRID:SCR_014284 | Patch clamp acquisition |

*Continued on next page*

*Continued*

| Reagent type (species) or resource | Designation | Source or reference | Identifiers | Additional information |
|---|---|---|---|---|
| Software, algorithm | Clampfit | Molecular Devices | RRID:SCR_011323 (pClamp) | Patch clamp analysis |
| Software, algorithm | Origin 2018 | OriginLab Corp | | Data fitting, figure preparation |
| Software, algorithm | MATLAB | MathWorks | RRID:SCR_001622 | Recovery from desensitization analysis |
| Software | VMD | https://www.ks.uiuc.edu/ Development/Download/ download.cgi?PackageName=VMD | RRID:SCR_001820 | Structure analysis, figure preparation, animation |
| Software, algorithm | PROPKA | https://github.com/ jensengroup/propka-3.1 | | pK$_a$ prediction |
| Software, algorithm | Modeller 9 v 20 | https://salilab.org/modeller/ download_installation.html | RRID:SCR_008395 | Protein structure modelling |
| Software, algorithm | CHARMM GUI | http://www.charmm-gui.org/ | | Constructing systems for simulation |

## Cell culture, mutagenesis and transfection

Human Embryonic Kidney 293 (HEK293) cells from ATCC (CRL-3216, lot 70005913) were used and identity confirmed using STR profiling. PCR based test for mycoplasma was last performed 7/2019 and was negative. HEK293 cells were maintained in Dulbecco's Modification of Eagle's Medium (DMEM) with 4.5 g/L glucose, L-glutamine and sodium pyruvate (Corning/Mediatech, Inc) or Minimum Essential Medium (MEM) with Glutamax and Earle's Salts (Gibco), supplemented with 10% FBS (Atlas Biologicals) and penicillin/streptomycin (Invitrogen). Cells were passaged every 2 to 3 days when approximately 90% confluence was achieved. HEK293 cells were plated on tissue culture treated 35 mm dishes, transfected 24 to 48 hr later and recorded from 24 to 48 hr post-transfection. Cells were transiently transfected using with chicken ASIC1 wild type or mutant and eGFP using an ASIC:eGFP ratio of 7.5:1 µg of cDNA per 10 mL of media. Transfections were performed using jet-PRIME (Polyplus Transfections) or polyethylenimine 25 k (PEI 25 k, Polysciences, Inc) following manufacturer's instructions, with media change at 6 to 8 hr. For non-stationary noise analysis, media was changed after 3–6 hr and recordings performed within 24 hr. Mutations were introduced using site-directed mutagenesis PCR and confirmed by sequencing (Fisher Scientific/Eurofins Genomics).

For experiments with non-canonical amino acid incorporation, HEK293 cells were co-transfected with three separate pcDNA3.1+ vectors each containing: (1) either wild type or L414TAG cASIC1, (2) R3 - two copies of orthogonal Bpa tRNA along with a single copy of the Bpa tRNA synthetase and (3) YAM – an additional copy of orthogonal tRNA at a mass ratio of 2:2:1, respectively. Our impression was that the addition of the YAM plasmid was not essential, but did seem to increase non-sense suppression efficiency. The tRNA and tRNA synthetase inserts were made by gene synthesis (Genescript, USA) using published sequences (*Ye et al., 2008*). Transfection was performed using PEI 25 k in a mass ratio of 1:3 (cDNA:PEI) for 6 to 8 hr, then the media was replaced with fresh supplemented MEM containing 40 µM MeO-Bpa, a methyl ester derivative of Bpa. Transfected cells were used for experiments 24–30 hr after the beginning of transfection.

## Electrophysiology and UV trapping

Culture dishes were visualized using a 20x objective mounted on a Nikon Ti2 microscope with phase contrast. A 470 nm LED (Thorlabs) and dichroic filter cube were used to excite GFP and detect transfected HEK cells. Outside-out patches were excised using heat-polished, thick-walled borosilicate glass pipettes of 3 to 15 MΩ resistance. Higher resistance pipettes were preferred for non-stationary noise analysis experiments. The pipette internal solution contained (in mM) 135 CsF, 33 CsOH, 11 EGTA, 10 HEPES, 2 MgCl$_2$ and 1 CaCl$_2$ (pH 7.4). External solutions with a pH greater than seven were composed of (in mM) 150 NaCl, 20 HEPES, 1 CaCl$_2$ and 1 MgCl$_2$ with pH values adjusted to their respective values using NaOH. For solutions with a pH lower than 7, HEPES was replaced with MES. All recordings were performed at room temperature with a holding potential of −60 mV using

an Axopatch 200B amplifier (Molecular Devices). Data were acquired using AxoGraph software (Axograph) at 20–50 kHz, filtered at 10 kHz and digitized using a USB-6343 DAQ (National Instruments). Series resistance was routinely compensated by 90% to 95% where the peak amplitude exceeded 100 pA. Rapid perfusion was performed using home-built, triple-barrel application pipettes (Vitrocom), manufactured according to *MacLean (2015)*. Translation of application pipettes was achieved using a piezo translator (P601.40 or P212.80, PI) mounted to a manual manipulator and driven by a voltage power supply (E505.00 or E-471.20, PI). Voltage commands to the piezo were first low-pass filtered (eight-pole Bessel; Frequency Devices) at 50–100 Hz. Solution exchange was routinely measured at the end of each patch recording using open tip currents with exchange times ranging from 250 to 500 µs.

For UV modulation, a high-power UV LED (KSL2-365, Rapp Optoelectronic) was used as the UV light source. The UV LED was set to maximum power and triggered by TTL input. The light emission was reflected off a 425 nm long-pass dichroic mirror held in a beam combiner (which combined the light from the 470 nm LED for GFP visualization), on through the epifluorescence port of the Ti2 microscope then reflected off of a 410 nm long-pass dichroic mirror before being focused onto the sample through a 20x objective. For resting state trapping experiments (*Figure 7*), a single sweep of UV involved 14 LED pulses of 50 ms in duration spaced by 450 ms, leading to a total of 700 ms exposure time spread across 7 s. For desensitized state trapping (*Figure 8*), baseline responses were recorded until rundown subsided and the peak amplitude stabilized. Subsequently, seven control responses were evoked followed by another seven responses with UV applications during pH 5 where the channels have fully desensitized, and finally seven additional post-UV responses. A single UV sweep was 20 LED pulses of 50 ms in duration spaced by 450 ms.

## Molecular dynamics simulations

Molecular dynamics simulations were performed using a structure of chicken ASIC1 suggested to be in the desensitized state (PDB code 4NYK, [*Gonzales et al., 2009*]), solved to a resolution of 3 Å. Residues 42–455 were resolved in the crystal structure. Of these residues, 23 had missing side chain atoms. The missing atoms were added using MODELLER 9v20 (*Sali and Blundell, 1993*), while the intracellular N- and C-termini were ignored. For each chain, the bound chloride ion and the 50 crystallographically resolved water molecules were retained. The initial membrane position was obtained from the Orientation of Proteins in Membranes database (*Lomize et al., 2012*). The simulated system, consisting of the protein, the chloride ions and crystallographic water molecules, embedded in a POPC lipid bilayer and surrounded by TIP3P water molecules and a NaCl concentration of 150 mM, was generated using the CHARMM GUI (*Jo et al., 2008*; *Lee et al., 2016*). Disulphide bonds for each chain were maintained between the following cysteine pairs: C94-C195, C173-C180, C291-C366, C309-C362, C313-C360, C322-C344 and C324-C336. The POPC bilayer was 120 Å x 120 Å, and the box length 146 Å. In the desensitized state, a number of acidic residues are believed to be protonated, however, exactly which residues is unclear. Since covalent bonds cannot be formed or broken during classical molecular dynamics simulations, the residues to protonate must be determined prior to performing simulations. Based on PROPKA $pK_a$ prediction (*Olsson et al., 2011*) using available ASIC structures in both open and desensitized states, as well as visual analysis of the structures, we chose to protonate most of the ionizable residues with a relatively consistent $pK_a$ above 5. This gave a protonation scheme in which two histidine residues and ten acidic residues were protonated, giving the following list of protonated residues: H74, E98, H111, E220, D238, E239, E243, E255, E314, E354, D408 and E417. All other side chains were retained in their standard protonation states. This low-pH protonation scheme yielded conformations which were adequately stable during simulations. For the simulations mimicking a higher pH value, all residues were kept in their standard ionization state (i.e. deprotonated for the acidic residues, neutral for histidine).

For simulations of the L414A mutant, the L414 side chain was manually changed to an alanine side chain prior to constructing the simulation systems.

The CHARMM36 force field was employed for proteins (*Best et al., 2012*) and lipids (*Klauda et al., 2010*), and the simulations were performed using GROMACS v 5.0.7 (*Abraham et al., 2015*). The systems were simulated in the NPT ensemble using periodic boundary conditions, and the equilibration protocol was as follows. The constructed systems were first energy minimized for 10,000 steps or until the maximum force acting on any atom was less than 1000 kJ mol$^{-1}$ nm$^{-1}$. This was followed by six shorter simulations, gradually releasing the position restraints

as suggested by the default CHARMM-GUI protocol. The first three short simulations were 25 ps long and used a time step of 1 fs; the fourth and the fifth were 100 ns long, while the final part of the equilibration was run for 2 ns. The equilibration simulations 4–6, as well as the production run, used a time step of 2 fs. In all steps, the Verlet cutoff scheme was used with a force-switch modifier starting at 10 Å and a cutoff of 12 Å. The cutoff for short-range electrostatics was 12 Å and the long-range electrostatics were accounted for using the particle mesh Ewald (PME) method (*Darden et al., 1993*; *Essmann et al., 1995*). The temperature was maintained at 310 K for all steps of the equilibration using a Berendsen thermostat (*Berendsen et al., 1984*), while the Nose-Hoover thermostat (*Hoover, 1985*; *Nosé, 1984*) was used to keep the temperature at 310 K for the production run. For the final four steps of the equilibration as well as for the production run, the pressure was maintained at 1 bar, using semi-isotropic pressure coupling. The Berendsen barostat (*Berendsen et al., 1984*) was employed for the equilibrations while the Parrinello-Rahman barostat (*Nosé and Klein, 1983*; *Parrinello and Rahman, 1981*) was used for the production run. Covalent bonds including hydrogen atoms were constrained using the LINCS algorithm (*Hess, 2008*). Snapshots were saved every 5 ps, and generally every fourth snapshot was used for analysis. Three repeats for each setup were performed (a, b and c), using different starting velocities for the first step of the equilibration. The simulation times were 3 × 200 ns for the protonated systems (wild type and L414A) and 3 × 400 ns for the deprotonated systems (wild type and L414A). Analysis was performed using standard tools in GROMACS as well as in-house tcl scripts run through VMD v 1.9.3 (*Humphrey et al., 1996*). Figures were prepared using VMD and Pymol (The PYMOL Molecular Graphics System, Version 2.0 Schrödinger, LLC).

## Kinetic simulations, statistics and data analysis

Kinetic simulations (*Figure 5—figure supplement 1*) were performed using Kinetic Model Builder in 'Eigen solver' mode (*Goldschen-Ohm et al., 2014*). Current desensitization decays were fitted using exponential decay functions in Clampfit (Molecular Devices). For recovery from desensitization experiments, the piezo command voltage was split and re-directed as an input signal. The resulting piezo 'mirror' signal was used to define conditioning and test pulse epochs. A custom script in Matlab (Mathworks) was used to detect peaks within each epoch and normalize the test pulse peak to the conditioning pulse. OriginLab (OriginLab Corp) was used to fit the normalized responses to:

$$I_t = \left(1 - e^{\left(\frac{-t}{\tau}\right)}\right)^m \tag{1}$$

Where $I_t$ is the fraction of the test peak at an interpulse interval of $t$ compared to the conditioning peak, $\tau$ is the time constant of recovery and $m$ is the slope of the recovery curve. Each protocol was performed between 1 and 3 times on a single patch, with the resulting test peak/conditioning peak ratios averaged together. Patches were individually fit and averages for the fits were reported in the text. N was taken to be a single patch.

For dose-response curves, patches were placed in the middle of a three-barrel application pipette and jumped to either side to activate channels with the indicated pH. Responses to higher pH values were interleaved with pH 5 applications on either side to control for any rundown. Peak currents within a patch were normalized to pH 5 and fit to:

$$I_x = \frac{1}{\left(1 + 10^{((pH_{50} - pH_x)n)}\right)} \tag{2}$$

where $I_x$ is the current at pH X, $pH_{50}$ is the pH yielding half maximal response and $n$ is the Hill slope. Patches were individually fit and averages for the fits were reported in the text. N was taken to be a single patch.

For non-stationary fluctuation analysis, runs of between 50 and 200 responses from a single patch were recorded. Within each recording, we identified the longest stretch of responses where the peak amplitude did not vary by more than 10%. We further eliminated individual traces with spurious variance such as brief electrical artifacts, resulting in blocks of 50–100 traces. To further correct for rundown or drift in baseline values we calculated the variance between successive traces, as opposed to calculating from the global average (*Heinemann and Conti, 1992*), using:

$$\delta_i^2 = \frac{\left(\frac{(T_{i+1}-T_i)}{2}\right)^2}{2} \tag{3}$$

where $\delta_i^2$ is the variance of trace $i$, $T_i$ is the current value of the trace $i$. The ensemble variance and current for each patch were divided into progressively larger time bins. The baseline variance was measured from a 50 ms time window just prior to pH 5 application. The resulting mean current-variance data were then fitted in Originlab using:

$$\sigma_i(I)^2 = iI - \frac{I^2}{N} + \left(\delta_{baseline}^2\right) \tag{4}$$

where $\sigma_i(I)^2$ is the variance, $i$ is the single channel current, $I$ is the average current, $N$ is the number of channels in the patch and $\sigma_{baseline}^2$ is the baseline variance. For all experiments, N was taken to be a single patch. Nonparametric two-tailed, unpaired randomization tests with 100,000 iterations were implemented in Python to assess statistical significance. Statistical comparisons of recovery from desensitization were based and reported on differences in recovery time constant.

## Acknowledgements

We thank Drs. Chris Ahern and Jason Galpin, University of Iowa, for conversion of Bpa to MeO-Bpa, as well as Lyndee Knowlton for technical assistance. Funding for this work was provided by NIH R00NS094761 and NARSAD Young Investigator Award to DMM, NSERC Discovery Grant RGPIN 2019–06864 and the Canada Research Chairs Program 950–232154 to MM, and Cystic Fibrosis Foundation Award LUECK18G0 to JDL, NIH T32GM068411-15 to MLR. WestGrid and Compute Canada are acknowledged for providing the computing facilities. This manuscript is dedicated to the memory of Dr. Jim R Howe.

## Additional information

### Funding

| Funder | Grant reference number | Author |
| --- | --- | --- |
| National Institute of Neurological Disorders and Stroke | R00NS094761 | David M Maclean |
| Brain and Behavior Research Foundation | NARSAD Young Investigator Award | David M Maclean |
| Natural Sciences and Engineering Research Council of Canada | RGPIN 2019-06864 | Maria Musgaard |
| Canada Research Chairs | 950-232154 | Maria Musgaard |
| Cystic Fibrosis Foundation | LUECK18G0 | John D Lueck |
| National Institute of General Medical Sciences | T32GM068411-15 | Matthew L Rook |

The funders had no role in study design, data collection and interpretation, or the decision to submit the work for publication.

### Author contributions

Matthew L Rook, Formal analysis, Investigation, Visualization; Abby Williamson, Software; John D Lueck, Resources, Funding acquisition, Methodology; Maria Musgaard, Formal analysis, Funding acquisition, Investigation, Visualization, Methodology; David M Maclean, Conceptualization, Formal analysis, Funding acquisition, Investigation, Visualization

### Author ORCIDs

Matthew L Rook (iD) http://orcid.org/0000-0002-5332-7678
Maria Musgaard (iD) http://orcid.org/0000-0001-6096-9014
David M Maclean (iD) https://orcid.org/0000-0001-8294-6075

### Decision letter and Author response

Decision letter https://doi.org/10.7554/eLife.51111.sa1
Author response https://doi.org/10.7554/eLife.51111.sa2

## Additional files

### Supplementary files

• Transparent reporting form

### Data availability

All data generated during this study are included in the manuscript and supporting files.

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
