## [Decision Letter]

**Acceptance summary:**

Desensitization of ligand-gated ion channels (LGICs) is a fundamental property that regulates the time course of their actions. However, the structural mechanisms that regulate entry and recovery from the desensitized state are still unclear for many LGICs. In this study, the authors use a powerful combination of approaches including ultra-fast solution exchange patch-clamp electrophysiology, MD simulations and unnatural amino acid photoactivable cross-linking to show that a change in conformation of Leu414 in the beta11-12 linker of the acid-sensing ion channel (ASIC) triggers ASIC desensitization. This paper uncovers significant new mechanistic information about ASIC desensitization kinetics.

**Decision letter after peer review:**

Thank you for submitting your article "β11-12 linker isomerization governs Acid-sensing ion channel desensitization and recovery" for consideration by *eLife*. Your article has been reviewed by three peer reviewers, including Cynthia M Czajkowski as the Reviewing Editor and Reviewer #1, and the evaluation has been overseen by Richard Aldrich as the Senior Editor. The following individuals involved in review of your submission have agreed to reveal their identity: Andrew J R Plested (Reviewer #2); Toshimitsu Kawate (Reviewer #3).

The reviewers have discussed the reviews with one another and the Reviewing Editor has drafted this decision to help you prepare a revised submission.

Summary:

Desensitization mechanism of ASICs remains poorly understood, despite the available crystal structures in the resting, open, and desensitized states. The recent study by Wu et al. suggested that Q277 (chicken numbering) enables channel desensitization through a mechanism explained by a steric effect. In contrast, the current paper by the MacLean group provides new experimental data that suggest isomerization of the β11-12 linker, rather than the steric effect by Q277, predominantly governs the transition between the closed and the desensitization states. The authors carefully characterized several mutants of an important residue in this linker (L414), using a combination of ultrafast-perfusion based patch clamp electrophysiology, non-stationary fluctuation analysis, and molecular dynamics simulations. All data support the idea that this residue flips outwardly during desensitization and the extent of energy barrier for this flip determines how quickly an ASIC channel desensitizes or recovers from desensitization. The authors also applied non-canonical amino acid cross-linking, which supports that the β11-12 linker rearrangement is necessary for channel desensitization.

This study is well-designed and the experiments are carefully performed. In particular, the fast perfusion system enabled the authors to demonstrate, for the first time, that a substitution of L414 with a smaller alanine residue does not affect the proton sensitivity. Furthermore, they were able to demonstrate that desensitization precedes a closed, rather than an open state. The reviewers agreed that these novel findings significantly improve our understanding of how ASICs desensitize.

Addressing the following essential revisions will strengthen the manuscript.

Essential revisions:

1) Need to disclose the protonation sites that were used in the MD studies. The reasoning behind what residues were chosen is also important and should be described. Without this information, there is no way for current or future scientists to replicate this work and to know if the simulation protonation states have any relation to the real situation.

2) Since recovery from desensitization is very sensitive to pH, some of the effects of the mutations in the β 11-12 linker might be due to changes in pK_a_ of the relevant residues for acid sensitivity of recovery that are altered by the mutations. Is the pH sensitivity of the recovery from desensitization substantially different between WT and L414A? This can be addressed by carrying out an experiment like in Figure 3—figure supplement 2 for the mutant channel. Alternatively, the mutant L414BzF after crosslinking should have lost pH sensitivity of recovery and this might be a way to confirm the mechanism. At the very least, the authors need to address and discuss this point thoroughly.

3) Surprisingly, authors did not observe any modulation of channel function when using desensitized state UV applications (data not shown). Authors should show the data even if the experimental protocol did not result in any observable effects. The authors need to discuss their ideas on why they did not see any effects especially in light of their MD simulations (Figure 4—figure supplement 1), showing multiple residues within 4 Å of L414 in the desensitized state.

4) The result that desensitization/recovery proceeds from closed channel states is an important conclusion from their data, which is only presented in the Results section. Additional discussion of this finding is warranted. Including a branched kinetic model that describes their data would highlight this finding but one can argue that it is not essential to support the main conclusions of this paper.

---

## [Author Response]

Essential revisions:1) Need to disclose the protonation sites that were used in the MD studies. The reasoning behind what residues were chosen is also important and should be described. Without this information, there is no way for current or future scientists to replicate this work and to know if the simulation protonation states have any relation to the real situation.

We report our protonation scheme in the first paragraph of the Materials and methods subsection “Molecular dynamics simulations”, and refer the reader to that section in the Results as well (subsection “L414A destabilizes the β11-12 linker upon deprotonation”, first paragraph). The following text in the Materials and methods describes our protonation scheme:

“In the desensitized state, a number of acidic residues are believed to be protonated, however, exactly which residues is unclear. […] For the simulations mimicking a higher pH value, all residues were kept in their standard ionization state (i.e., deprotonated for the acidic residues, neutral for histidine).”

We acknowledge that the used protonation scheme may not be the correct representation of the physiological state. However, the purpose of protonating residues in this work was to stabilize the channel in the proposed desensitized state, which our chosen protonation scheme does do. Indeed, we observed greater structural changes upon deprotonation (Figure 3, Figure 4—figure supplement 1).

2) Since recovery from desensitization is very sensitive to pH, some of the effects of the mutations in the β 11-12 linker might be due to changes in pK_a_ of the relevant residues for acid sensitivity of recovery that are altered by the mutations. Is the pH sensitivity of the recovery from desensitization substantially different between WT and L414A? This can be addressed by carrying out an experiment like in Figure 3—figure supplement 2 for the mutant channel. Alternatively, the mutant L414BzF after crosslinking should have lost pH sensitivity of recovery and this might be a way to confirm the mechanism. At the very least, the authors need to address and discuss this point thoroughly.

We have carried out the proposed L414A recovery experiment at different inter-pulse pH values and added the data as a new supplementary figure (Figure 3—figure supplement 3). We find that L414A is fast at all the pH’s we examined, with small although detectable changes. That is, the pH dependence of recovery is retained but the effect is smaller over the range we examined. This observation suggests that the 414 position plays a key role in the recovery process and does not simply produce apparent changes in recovery by shifting the pH-dependence of recovery slightly to the right or left. Furthermore, it is unlikely that converting a non-polar Leu to a non-polar Ala in a region with few charged residues could produce the local electrostatic change required to appreciably alter pK_a_ values of critical residues in the surrounding area. Therefore we favor the interpretation that L414A works predominantly by accelerating the recovery transition, and not by shifting its pH-dependence. We have included these considerations in the Results (subsection “L414A destabilizes the β11-12 linker upon deprotonation”, last paragraph).

3) Surprisingly, authors did not observe any modulation of channel function when using desensitized state UV applications (data not shown). Authors should show the data even if the experimental protocol did not result in any observable effects. The authors need to discuss their ideas on why they did not see any effects especially in light of their MD simulations (Figure 4—figure supplement 1), showing multiple residues within 4 Å of L414 in the desensitized state.

We were also surprised by this. In our original desensitized state dataset, rundown throughout our recordings of small amplitude signals masked any UV-correlated peak inhibition. With more experience in UV-trapping and ncAA incorporation, we have now gone back, modified the protocol and found evidence of trapping. We specifically adjusted our approach in two ways. First, after whole cell recording was established we performed a variable number of pH applications until the peak response stabilized. This essentially allowed rundown/tachyphylaxis to occur after which we began the experiments depicted in Figure 8 (seven control applications, followed by seven UV applications and another seven control). Second, we repeated the experiment using greater total UV dose (seven trains of 20x 50 ms pulses as opposed to five trains of 14x 50 ms in the resting state data). While this more aggressive irradiation protocol was harder on the cells, it did reveal state-dependent inhibition of the peak response (~ 50%) as expected if the channels were trapped in the desensitized state. We have now included this new data as an additional figure (Figure 8) with new Materials and methods (subsection “Electrophysiology and UV trapping”, last paragraph), Results (subsection “ncAA incorporation and UV-trapping at Leu414”, last paragraph) and Discussion (subsection “Conclusion”).

4) The result that desensitization/recovery proceeds from closed channel states is an important conclusion from their data, which is only presented in the Results section. Additional discussion of this finding is warranted. Including a branched kinetic model that describes their data would highlight this finding but one can argue that it is not essential to support the main conclusions of this paper.

We thank the reviewers for their interest on this point. However, we did not wish to convey that we *conclude* desensitization/recovery precedes exclusively from closed states. Only that we have found some evidence which favors that hypothesis over the idea that transitions occur primarily to/from open states. Diagnosing the procession of states is challenging (Lin and Stevens, 1994; Colquhoun and Hawes, 1995) and we feel more evidence would be needed before we can make some conclusion. In light of this, we have moved one section of the Results (discussing how linear reactions predict slow deactivation/resurgent current when recovery is fast) into the Discussion (subsection “Does ASIC desensitization proceed from open or shut states?”) and added a note to emphasize more experiments would be needed to support the shut state hypothesis.

With regard to kinetic models, prior to submission we did some branched versus linear reaction scheme simulations of desensitization and recovery kinetics for wild type and L414A-like channels. These simulations quickly reveal how grossly inadequate simple models are for ASIC behavior. In either model if we try to approximate L414A kinetics by increasing the rate constant for recovery from desensitization, a robust increase in steady-state/equilibrium current follows (from 1% of peak to >70% of peak). This is not reflected by the data. We hypothesize that this discrepancy in equilibrium current amplitude between model and data arises because of the pH dependence of recovery. As seen in Figure 3—figure supplement 2, recovery gets slower and slower with acidic interpulse pH values. If one extrapolates this trend then the recovery route at pH 7 or 6.5 or pH 5 would be exceedingly unfavorable, resulting in the minimal steady-state current observed. We have included this notion and one such very simple 4-state branching model in a new Discussion section (subsection “The recovery from desensitization process”), aimed at highlighting this and other ideas which emerged as a consequence of this review.